# *Azospirillum brasilense* as a Bioinoculant to Alleviate the Effects of Salinity on Quinoa Seed Germination

**DOI:** 10.3390/plants14243829

**Published:** 2025-12-16

**Authors:** Jose David Apaza-Calcina, Milagros Ninoska Munoz-Salas, Flavio Lozano-Isla, Rachel Passos Rezende, Raner José Santana Silva

**Affiliations:** 1Laboratório de Biotecnologia Microbiana, Centro de Biotecnologia e Genética, Campus Soane Nazaré de Andrade, Universidade Estadual de Santa Cruz, Ilhéus 45662-900, Bahia, Brazil; rachel@uesc.br; 2Dirección de Recursos Genéticos y Biotecnología, Estación Experimental Agraria Perla del Vraem, Instituto Nacional de Innovación Agraria (INIA), Cusco 00800, Peru; 3Department of Earth and Environment, Institute of Environment, Florida International University, Miami, FL 33199, USA; mmuno126@fiu.edu; 4Facultad de Ingeniería y Ciencias Agrarias, Universidad Nacional Toribio Rodríguez de Mendoza de Amazonas (UNTRM), Chachapoyas 1001, Peru; flavio.lozano@untrm.edu.pe; 5Dirección de Supervisión y Monitoreo en las Estaciones Experimentales, Estación Experimental Agraria El Chira, Instituto Nacional de Innovación Agraria (INIA), Piura 20120, Peru; 6Programa de Pós-Graduação em Genética e Biologia Molecular, Campus Soane Nazaré de Andrade, Universidade Estadual de Santa Cruz, Ilhéus 45662-900, Bahia, Brazil

**Keywords:** plant growth-promoting bacteria (PGPB), salinity stress, seedling growth, sodium chloride, sustainable agriculture

## Abstract

Quinoa (*Chenopodium quinoa* Willd.) is valued for its resilience to abiotic stress; however, germination and seedling establishment remain highly sensitive to salinity. While its salt tolerance at later growth stages has been well studied, strategies to improve early development under high salinity are limited, and the role of halotolerant plant growth-promoting bacteria (PGPB) in quinoa has not been systematically investigated. This study assessed the ability of three *Azospirillum brasilense* strains (BR-11001, BR-11002, and BR-11005) to increase the germination and seedling performance of the cultivar ‘BRS Piabiru’ under saline stress. A 3 × 4 factorial design with three bacterial treatments and four NaCl concentrations (0, 150, 300, and 450 mM) was conducted in a completely randomized arrangement, with four replicates per treatment. Seeds were surface sterilized, inoculated, and incubated at 18 °C under constant light for 10 days. Elevated salinity (≥300 mM NaCl) drastically reduced germination and seedling vigor in the controls. Inoculation with BR-11002 significantly alleviated salinity-induced damage, sustaining over 84% germination at 450 mM and increasing seedling biomass at 300 mM. These findings highlight the potential of halotolerant *A. brasilense*, particularly BR-11002, as bioinoculants to promote quinoa establishment in salt-affected soils, supporting sustainable agriculture and food system resilience.

## 1. Introduction

Salinity has emerged as a pressing global challenge that increasingly threatens agricultural sustainability. Estimates from the Food and Agriculture Organization [1] indicate that more than 900 million hectares of land worldwide are affected by salt accumulation, including nearly one-fifth of all irrigated land. This widespread salinization is largely driven by poor irrigation management, limited drainage infrastructure, and intensifying climate variability, particularly in arid and semiarid ecosystems where evapotranspiration exceeds precipitation rates [2]. Salinity stress disrupts plant physiological processes through osmotic imbalance, ion toxicity—primarily from Na^+^ and Cl^−^—and the excessive generation of reactive oxygen species (ROS), which collectively impair germination, limit root growth, and hinder crop performance [3].

Quinoa (*Chenopodium quinoa* Willd.) has garnered increasing attention as a climate-resilient crop because of its high nutritional quality and ability to tolerate adverse agroecological conditions [4]. Although quinoa demonstrates moderate salt tolerance during vegetative growth, its early developmental stages, particularly seed germination and seedling establishment, are highly sensitive to elevated NaCl levels [5]. Delayed emergence and uneven seedling vigor are common in saline soils, thereby limiting crop stand uniformity and yield potential [6]. Increasing salt tolerance during these critical early phases is essential for enabling quinoa cultivation in salt-affected marginal environments.

In this context, plant growth-promoting bacteria (PGPB) represent a promising and environmentally sustainable strategy to support crop establishment under saline conditions [7]. PGPB contributes to plant resilience through diverse mechanisms [8], including biological nitrogen fixation, phosphate solubilization, siderophore production, and the synthesis of growth-promoting hormones such as indole-3-acetic acid (IAA). They also activate antioxidant defense systems that mitigate ROS-induced oxidative damage [7].

Within the diverse group of plant growth-promoting bacteria (PGPB), several genera are well recognized for their beneficial roles in agriculture. *Rhizobium* and *Bradyrhizobium* enhance soil fertility through symbiotic nitrogen fixation in legumes, whereas *Pseudomonas* spp. efficiently colonize the rhizosphere and act as potent biocontrol agents by producing siderophores and antifungal metabolites. *Bacillus* spp., particularly *B. subtilis*, are notable for their ability to form stress-resistant endospores, synthesize phytohormones, and induce systemic resistance in host plants. Additional PGPB, such as *Enterobacter*, *Klebsiella*, and *Burkholderia*, further contribute to auxin biosynthesis, nutrient solubilization, and improved tolerance to abiotic stresses. This taxonomic diversity underscores the broad functional repertoire of PGPB in enhancing crop resilience under challenging environmental conditions [9].

Among these, *Azospirillum brasilense* stands out as one of the most extensively studied species because of its ability to promote root development, nutrient uptake, and biomass accumulation under abiotic stress in nonleguminous species [10,11,12]. A central mechanism underlying its plant growth-promoting activity is the production of phytohormones, most notably indole-3-acetic acid (IAA). The ability of *A. brasilense* to stimulate root proliferation and increase stress tolerance is strongly linked to its ability to synthesize IAA via tryptophan-dependent metabolic pathways. The indole-3-pyruvic acid (IPyA) pathway is considered the predominant route, involving the transamination of tryptophan to indole-3-pyruvate, decarboxylation to indole-3-acetaldehyde, and subsequent oxidation to IAA. Alternative pathways—including the indole-3-acetamide (IAM), tryptamine (TAM), and indole-3-acetonitrile (IAN) pathways—may also operate under specific environmental conditions, providing metabolic flexibility in IAA biosynthesis [13,14]. This biochemical versatility explains the broad and consistent growth-promoting effects of *Azospirillum* across diverse plant species and stress environments.

Recent efforts have focused on halotolerant and extremophilic bacterial strains isolated from saline soils, which exhibit enhanced potential to confer salt tolerance in crops such as wheat, rice, and soybean [9,10]. These microorganisms promote seedling vigor and germination under salinity by improving ion homeostasis, limiting lipid peroxidation, and increasing antioxidant enzyme activity [11]. Nevertheless, few studies have examined the effects of *A. brasilense* strains on quinoa performance under saline stress, leaving a gap in understanding their potential applications in quinoa-based cropping systems.

To address this gap, the present study was designed to investigate the role of halotolerant *Azospirillum brasilense* strains in mitigating salinity stress during the germination and early growth stages of *Chenopodium quinoa*. Although these bacteria have been studied in cereals and other nonleguminous crops, their application in quinoa remains unexplored. This research is the first screening of halotolerant *A. brasilense* strains for their ability to improve germination, seedling establishment, and physiological performance under saline conditions. Specifically, the study pursued four objectives: (i) to evaluate the growth potential of six bacterial strains in saline and nonsaline media to assess viability and halotolerance; (ii) to assess the impact of a NaCl concentration gradient (0–450 mM) on seed germination and cotyledon emergence as indicators of inoculation efficacy; (iii) to determine the influence of inoculation on seedling growth parameters—including root and shoot length and biomass—under saline stress; and (iv) to examine physiological stress responses by quantifying the activity of key antioxidant enzymes.

## 2. Results

### 2.1. Bacterial Growth Under Saline and Nonsaline Conditions

Bacterial viability, assessed through the optical density at 600 nm (OD_600_), was significantly affected by strain identity, salinity level, and their interaction (*p* < 0.001, Figure 1). Across the salinity gradient (0–900 mM NaCl), all the *A. brasilense* strains presented progressive reductions in the OD_600_, with statistically significant differences among the strains at each salt concentration (*p* < 0.05).

At 0 mM NaCl, BR-11002 presented the highest OD_600_ (1.4700), followed by BR-11001 (1.2925) and BR-11004 (1.1825). BR-11003 presented the lowest growth (0.9525). As the salinity increased to 150 mM, BR-11002 and BR-11001 maintained OD_600_ values above 1.28, whereas BR-11003, BR-11005, and BR-11006 decreased below 1.16. At 300 mM NaCl, BR-11002 (1.2500) and BR-11001 (1.1250) sustained significantly greater growth than the other strains did (*p* < 0.05), with the OD_600_ values for BR-11005 and BR-11006 decreasing below 1.00.

At 450 mM, BR-11002 (1.0025) and BR-11001 (0.8825) retained OD_600_ values closest to 1.0, whereas BR-11003 and BR-11006 were reduced to 0.6125 and 0.9275, respectively. Growth inhibition intensified at 600 mM and 750 mM, with BR-11002 and BR-11001 maintaining the highest OD_600_ values (0.7875 and 0.7000 at 600 mM and 0.7025 and 0.6000 at 750 mM, respectively). At 900 mM, all the strains presented substantial reductions in the OD_600_, yet the values of BR-11002 (0.5625) and BR-11001 (0.4825) remained significantly greater than those of BR-11003 (0.1750), BR-11004 (0.3675), and BR-11006 (0.2650) (*p* < 0.001). The full factorial analysis indicated a significant strain × NaCl interaction (F_6,121_ = 18.121, *p* < 0.001), confirming differential salinity tolerance among the strains across NaCl levels.

### 2.2. Germination Dynamics and Cotyledon Emergence Under Salinity Stress

The germination behavior of *Chenopodium quinoa* was significantly influenced by both inoculation with *Azospirillum brasilense* and increasing NaCl concentrations. Two-way ANOVA revealed highly significant main effects of strain, salinity level, and their interaction on all germination-related variables (*p* < 0.001, Figure 2). Concentrations above 300 mM NaCl (e.g., 450 mM) drastically inhibited germination, resulting in seed necrosis and severe seedling deformation, whereas 300 mM NaCl still allowed a measurable percentage of germination with clear differences in seedling vigor compared with the control.

The germination percentage remained unaffected by salinity up to 150 mM NaCl across all the treatments, maintaining values of 100%. At 300 mM, a minor reduction was observed in the uninoculated control (95.5%), whereas BR-11001 and BR-11002 maintained high germination rates (98.5% and 99.0%, respectively). However, under 450 mM NaCl, germination was drastically reduced in the control to 52.5%, whereas BR-11001 and BR-11002 significantly mitigated this effect, maintaining germination at 78.5% and 84.0%, respectively (Figure 3a).

The mean germination time (MGT) increased proportionally with salinity across all the treatments. At 0 mM NaCl, MGT was shortest in BR-11001 (1.15 days) and BR-11002 (1.11 days) compared with the control (1.36 days). Under 450 mM NaCl, MGT was significantly prolonged in all the treatments but remained lower in the inoculated seeds: 5.74 days in the control versus 3.53 and 3.64 days in BR-11001 and BR-11002, respectively (Figure 3b).

The germination uncertainty (GU) increased steadily with increasing salinity. At 0 mM NaCl, BR-11001 and BR-11002 presented the lowest GU values (0.53 and 0.50), whereas the control presented a greater dispersion (0.77). At 450 mM, the uncertainty increased to 2.91 for BR-11001, 2.83 for BR-11002, and 2.86 for the control, indicating reduced uniformity of germination events under high salinity (Figure 3c).

The germination synchrony (GS) followed an inverse trend, decreasing as the NaCl concentration increased. Under nonsaline conditions, synchrony was highest in BR-11001 (0.83), followed by BR-11002 (0.80) and the control (0.75). At 450 mM, GS decreased across all the treatments but remained significantly greater in the inoculated treatments (0.13 and 0.15 for BR-11001 and BR-11002, respectively) than in the control (0.12) (Figure 3d).

### 2.3. Seedling Traits and Biomass Allocation

Significant main effects of inoculation, salinity level, and their interaction (*p* < 0.001) were detected for all measured seedling parameters, including shoot length, root length, dry weight, and the seedling vigor index (SVI) (Figure 4). Compared with the uninoculated control, inoculation with the *A. brasilense* strains BR-11001 and BR-11002 consistently enhanced seedling performance across all salinity treatments.

Shoot length progressively decreased with increasing NaCl concentration, but the extent of the decrease was substantially mitigated by bacterial inoculation. Under nonsaline conditions (0 mM), the shoot length reached 6.03 cm and 5.81 cm in BR-11002 and BR-11001, respectively, whereas it was 4.77 cm in the control. At 450 mM NaCl, BR-11002 maintained a significantly greater shoot length (3.96 cm) than did BR-11001 (3.06 cm) and the control (2.11 cm) (Figure 4a).

The length of the roots followed a similar pattern. At 0 mM, BR-11002 had the highest value (3.93 cm), followed by BR-11001 (3.32 cm) and the control (3.05 cm). As the salinity increased to 450 mM, the degree of root elongation decreased in all the treatments, with values of 2.54 cm (BR-11002), 2.10 cm (BR-11001), and 0.93 cm (control) (Figure 4b).

Dry biomass accumulation was also significantly influenced by both salinity and inoculation. At 0 mM, the BR-11002-treated seedlings presented the greatest dry weight (0.049 g), outperforming the BR-11001-treated (0.032 g) and control (0.026 g) plants. Even under severe salt stress (450 mM), BR-11002 maintained a greater biomass yield (0.0316 g), whereas the biomass yield of BR-11001 and the control decreased to 0.0258 g and 0.0185 g, respectively (Figure 4c).

The seedling vigor index (SVI) sharply decreased with increasing salinity, but the inoculated plants presented relatively high values. At 0 and 150 mM NaCl, the SVI remained statistically unchanged between BR-11002 (~995–1001) and BR-11001 (~909–912), whereas the control had significantly lower scores (~772–782). At 450 mM, the SVI decreased to 546.2 (BR-11002), 405.1 (BR-11001), and 159.8 (control) (Figure 4d).

### 2.4. Antioxidant Enzyme Activities in Response to Salt Stress and Inoculation

The activities of all four antioxidant enzymes—superoxide dismutase (SOD), catalase (CAT), ascorbate peroxidase (APX), and guaiacol peroxidase (GPX)—were significantly affected by salinity, *A. brasilense* inoculation, and their interaction (*p* < 0.001, Figure 5).

SOD activity increased with increasing NaCl concentration in all the treatments. At 0 mM, the values were 23.96 U min^−1^ mg^−1^ protein in the control, 26.78 in BR-11001, and 26.67 in BR-11002. At 450 mM, the SOD activity reached 56.25 in the control, 63.36 in BR-11001, and peaked at 68.12 U min^−1^ mg^−1^ protein in BR-11002 (Figure 5a).

CAT activity also tended to increase with increasing salinity. At 0 mM NaCl, the CAT activity values were 14.15 (control), 14.94 (BR-11001), and 16.01 (BR-11002). Under 450 mM, the CAT activity increased to 35.73 in the control, 41.70 in BR-11001, and 46.06 U min^−1^ mg^−1^ protein in BR-11002 (Figure 5b).

The APX activity followed a similar pattern. At 0 mM, the control had a value of 9.36, whereas the corresponding values for BR-11001 and BR-11002 were 10.84 and 10.76, respectively. At 450 mM, the APX values reached 25.59 (control), 29.64 (BR-11001), and 33.05 U min^−1^ mg^−1^ protein (BR-11002) (Figure 5c).

GPX activity was lowest under nonsaline conditions, with values of 7.15 in the control, 7.93 in BR-11001, and 7.62 in BR-11002. At 450 mM NaCl, GPX activity increased to 17.78 (control), 20.86 (BR-11001), and 23.38 U min^−1^ mg^−1^ protein in BR-11002 (Figure 5d).

### 2.5. Multivariate Analysis of Phenotypic and Biochemical Responses Under Salinity Stress

To understand the interaction effects of phenotypic and biochemical traits on salinity tolerance in quinoa, a principal component analysis (PCA) was conducted (Figure 6).

The first two principal components explained 95.9% of the total variance, with Dim 1 accounting for 74.3% and Dim 2 accounting for 21.6%. Phenotypic traits, including shoot length (SL), root length (RL), dry weight (DW), and the seedling vigor index (SVI), were strongly associated with positive values on Dim 1, whereas biochemical traits—superoxide dismutase (SOD), catalase (CAT), ascorbate peroxidase (APX), and guaiacol peroxidase (GPX)—clustered together and loaded in opposite directions along this dimension (Figure 6A).

The distribution of individuals further revealed clear separation among the treatments (Figure 6B). Seedlings inoculated with BR-11002 were grouped in the upper-right quadrant across all NaCl concentrations, reflecting enhanced performance under saline conditions. In contrast, the BR-11001 treatments were positioned in intermediate zones, showing partial improvement compared with the controls but lower than that of BR-11002. The controls clustered in the lower-right to lower-left quadrants, particularly at relatively high salinity levels (300–450 mM), indicating reduced growth and stress tolerance.

## 3. Discussion

The present study underscores the potential of *Azospirillum brasilense*, particularly strain BR-11002, to improve early-stage salinity tolerance in *Chenopodium quinoa* through a multifaceted suite of mechanisms, including enhanced germination, seedling growth, and antioxidant responses. These results support the emerging role of halotolerant plant growth-promoting bacteria (PGPB) in the development of sustainable strategies to mitigate abiotic stress in crops [15,16]. Recent studies have confirmed that inoculation with halotolerant PGPB can alleviate salt-induced damage by modulating hormonal signaling, enhancing ion and nutrient uptake, and activating antioxidative pathways [15,17,18].

### 3.1. Germination Dynamics Under Saline Conditions

The germination dynamics were significantly stabilized under saline conditions by bacterial inoculation. Seeds treated with BR-11002 maintained germination rates above 84% at 450 mM NaCl, whereas the germination rate of the uninoculated control decreased to 52.5%. This marked increase under osmotic stress reflects the capacity of *A. brasilense* to facilitate early water uptake, protect membrane integrity, and maintain cellular turgor, mechanisms also observed in cereals and legumes inoculated with PGPB [19,20,21,22]. Moreover, the reduced mean germination time (MGT) and germination uncertainty (GU), along with increased synchrony, suggest that inoculated seeds benefit from a more coordinated and efficient germination process, which is crucial for uniform field emergence under stress-prone conditions.

The beneficial effects of *A. brasilense* during germination under nonsaline conditions were reported by Brito et al. [23], who reported that the inoculation of quinoa seeds with *A. brasilense* increased germination by up to 17%, further supporting the role of PGPB in enhancing establishment and vigor during early growth. Other studies have demonstrated that coinoculation with halotolerant PGPB can further increase quinoa performance under salinity stress. Yang et al. demonstrated that inoculation with halotolerant strains of *Enterobacter* sp. and *Bacillus* sp. significantly improved early seedling growth and antioxidant enzyme activity in quinoa under saline conditions, highlighting the value of integrating multiple microbial partners for increased tolerance. Similarly, Yang et al. [24] reported that inoculation with *Burkholderia phytofirmans* PsJN stimulated the growth, yield, and physiological performance of quinoa grown under salinity stress, underscoring the effectiveness of consortia or complementary strains in improving both early and late developmental stages. In addition, Aslam et al. [25] reported that inoculation with drought-tolerant rhizobacteria significantly enhanced quinoa growth and physiological traits under water deficit, suggesting that microbial inoculation is a versatile approach for improving stress resistance across multiple abiotic stresses.

### 3.2. Seedling Responses and Enzyme Activity

In addition to germination, seedling development also markedly improved. The inoculated seedlings presented significantly longer shoot and root lengths, greater dry biomass, and greater seedling vigor indices across all salinity levels. These findings are consistent with previous reports demonstrating that *A. brasilense* promotes vegetative growth through phytohormone production (especially IAA production), improves ion homeostasis, and stimulates lateral root development [26,27,28,29]. Such enhancements contribute to greater resource acquisition and physiological stability, giving seedlings a competitive advantage during the critical early stages of establishment under saline conditions.

The biochemical response further confirmed the functional role of *A. brasilense* in stress mitigation. Inoculated seedlings, particularly those treated with BR-11002, presented significantly elevated activity of key antioxidant enzymes, such as SOD, CAT, APX, and GPX, especially at 300 and 450 mM NaCl. These enzymes constitute the core of the ROS-detoxifying system and are essential for preserving membrane integrity and protein function during abiotic stress [30,31,32]. Comparable enzymatic enhancement has been observed in soybean and tomato plants inoculated with stress-tolerant bacteria, where improved antioxidant profiles were linked to reduced lipid peroxidation and better photosynthetic performance [33,34,35,36].

Additionally, the integration of phenotypic and biochemical traits through PCA revealed a distinct trade-off between growth parameters and antioxidant enzyme activities, positioned in opposite directions along the first principal component (Figure 6). This distribution suggests that under salinity stress, plants must allocate resources between sustaining growth and activating antioxidant defenses. BR-11002-inoculated plants presented elevated values for both growth-related traits and enzymatic protection, highlighting their superior capacity to maintain seedling performance across salt stress conditions.

### 3.3. Insights and Limitations

These physiological and biochemical responses are likely governed by a complex interplay between microbial traits and plant regulatory pathways. Beneficial rhizobacteria such as *A. brasilense* are known to synthesize exopolysaccharides, osmoprotectants, and hormones that prime plants for stress resistance. In parallel, host plants activate transcriptional responses that reinforce antioxidant defenses and facilitate ion compartmentalization. Transcriptomic analyses in rice and wheat have demonstrated that *A. brasilense* inoculation upregulates genes involved in ABA signaling, redox homeostasis, and ion transport under salt stress [15,37,38]. For example, Reference [37] reported that ABA accumulation under salinity affects root meristem activity and cell expansion, a process modulated by bacterial inoculation through hormonal crosstalk. Although gene expression was not directly assessed in the present study, the observed phenotypic and biochemical responses strongly suggest that analogous regulatory mechanisms were activated in quinoa.

The present study focused mainly on germination and biochemical responses during the early development of quinoa via the PGPB *Azospirillum brasilense* under salinity stress, without assessing gene expression or molecular mechanisms. This limits the direct linkage of the observed changes to regulatory pathways. Future research should integrate molecular approaches and evaluate consortium inoculation systems.

### 3.4. Implications for Sustainable Agriculture

In light of the increasing prevalence of salt-affected soils and the increasing global interest in quinoa as a resilient and nutritious crop, the use of the *A. brasilense* strain BR-11002 as a bioinoculant represents a promising and eco-compatible strategy. The consistent improvements in germination, seedling vigor, and antioxidant capacity highlight the versatility and potential scalability of the strain in saline agroecosystems. These findings support the integration of halotolerant PGPB into sustainable agricultural frameworks aimed at enhancing crop performance in degraded and marginal lands.

## 4. Materials and Methods

### 4.1. Biological Materials and Bacterial Strains

A total of six *Azospirillum brasilense* strains (BR-11001, BR-11002, BR-11003, BR-11004, BR-11005, and BR-11006) were provided by the Brazilian Agricultural Research Corporation (EMBRAPA) (Brasilia, Brazil). Among them, three strains (BR-11001, BR-11002, and BR-11005) have been characterized previously in studies addressing their taxonomic position, physiological traits, and effects on plant growth.

BR-11001 (Sp7), which was originally isolated from *Digitaria decumbens* [39], has been used as a reference strain in studies of associative diazotrophic bacteria because of its ability to fix nitrogen, synthesize auxins, promote plant growth, and improve maize seedling development and biomass accumulation [40] as well as productivity in *Gaillardia pulchella* when combined with nitrogen fertilization [41]; moreover, it has been employed as a control for evaluating new inoculants [42]. The strain BR-11002 (SpCd), which was isolated from the rhizosphere of *Cynodon dactylon* in the United States, has also been investigated as a plant growth-promoting rhizobacterium (PGPR) because of its ability to fix nitrogen and synthesize phytohormones such as auxins, cytokinins, and gibberellins; it has served as a comparative model in studies of root colonization and diazotrophy in grasses [43], and genome-based analyses confirmed its close relationship with *A. brasilense* Sp7 while revealing distinct genomic signatures that highlight the intraspecific diversity of this species complex [44]. Finally, BR-11005 (Sp245), which is isolated from wheat roots in southern Brazil, has been recognized for its ability to mitigate abiotic stress, as maize seed inoculation with this strain improved water status, proline accumulation, and biomass production under drought [45], whereas more recent studies in sorghum have demonstrated its contribution to dry matter accumulation, nutrient uptake, and tolerance to water and nitrogen stress in semiarid environments [46], reinforcing its potential as a bioinoculant in sustainable agriculture.

Seeds of *Chenopodium quinoa* Willd. The cultivar ‘BRS Piabiru’ was obtained from the EMBRAPA germplasm collection. This genotype is adapted to tropical and subtropical climates and is recognized for its early maturation, high productivity, and resilience to biotic and abiotic stresses [47]. Furthermore, it has a high phenolic content, including quercetin and kaempferol derivatives, contributing to its antioxidant potential and stress tolerance [48]. This cultivar was chosen to assess physiological responses to salt stress and evaluate the potential benefits of microbial inoculation.

### 4.2. Evaluation of Salinity Tolerance in A. brasilense Strains

To assess halotolerance, each of the six *A. brasilense* strains was cultured in Luria–Bertani (LB) broth containing a gradient of sodium chloride concentrations (0, 150, 300, 450, 600, 750, and 900 mM NaCl). The initial inocula were standardized to an optical density (OD_600_) of 0.1, incubated in 10 mL of LB medium at 35 °C for 48 h, and shaken at 200 rpm. Bacterial growth was quantified spectrophotometrically by recording the optical density at 600 nm (OD_600_) via a Rayleigh UV-Vis spectrophotometer (Rayleigh Instruments, Shanghai, China). Additionally, bacterial viability was estimated by serial dilution and plating on nutrient agar supplemented with corresponding NaCl concentrations. Colony-forming units (CFUs) were enumerated after 48 h of incubation. All the experiments were conducted in triplicate.

### 4.3. Inoculum Preparation and Seed Treatment

Following the salinity assay, the *A. brasiliense* strains BR-11001 (rhizospheric) and BR-11002 (endophytic) were selected for further experiments because of their superior growth performance under saline conditions. These strains were individually cultured in 250 mL of LB broth at 35 °C for 24 h on an orbital shaker (PsycroTherm, New Brunswick Scientific, New Brunswick, NJ, USA) at 120 rpm. Bacterial cells were harvested by centrifugation (5000× *g*, 10 min) and resuspended in sterile distilled water. The final suspension was adjusted to an OD_600_ = 0.400, corresponding to an approximate density of 10^8^ CFU/mL.

Quinoa seeds were disinfected by immersion in 75% ethanol for 1 min, followed by treatment with 2% sodium hypochlorite for 2 min. Seeds were then rinsed thoroughly three times with sterile distilled water. For inoculation, sterilized seeds were soaked in the bacterial suspension for 45 min at 25 °C, following the methodology outlined by Barbieri et al. [49].

### 4.4. Germination Assay Under Salinity Stress

The germination experiment was structured as a completely randomized block design (CRBD) within a 3 × 4 factorial framework, incorporating three inoculation treatments—*A. brasilense* BR-11001, BR-11002, and a noninoculated control—and four sodium chloride (NaCl) concentrations—0, 150, 300, and 450 mM. This resulted in 12 treatment combinations, each replicated four times, with 50 seeds per replicate. Seeds were sown on germination paper moistened with NaCl solution at a volume equal to 2.5 times the dry weight of the paper and enclosed in seed germination boxes [50,51]. The germination process was carried out in a growth chamber set at 18 °C for a period of 10 days. On the fourth day, rehydration was performed using the respective saline solution.

Daily observations were made to monitor radicle emergence, with germination considered effective when the radicle extended at least 2 mm, following the International Rules for Seed Testing [52]. To quantify germination performance and seed vigor, the following indices were calculated: germination percentage (GP), mean germination time (MGT), synchronization index (SYN), and uncertainty index (UNC). These metrics were derived via the equations described by Ranal and Santana [53] and implemented in the *GerminaR 2.1.5* R package [54].

### 4.5. Seedling Growth Evaluation

Seedling morphometric measurements were collected 10 days after sowing to assess early-stage biomass accumulation and physiological performance under saline stress conditions. Shoot and root lengths were recorded in centimeters, whereas dry biomass was determined in grams after drying the seedlings in a forced-air oven at 65 °C until a constant weight was achieved. The seedling vigor index (SVI) was used as an integrative metric to assess early plant development under stress conditions by combining seed germination capacity and seedling growth performance [55]. The SVI was calculated via the following formula:$$\mathrm{SVI}=\left(\bar{L_{r}}+\bar{L_{s}}\right)\times \bar{GP}$$
where $\bar{L_{r}}$ is the mean root length, expressed in centimeters (cm), and where $\bar{L_{s}}$ is the mean shoot length, also expressed in centimeters. The term $\bar{GP}$ represents the average germination percentage of the seeds, expressed as a percentage (%).

### 4.6. Antioxidant Enzyme Activity Assays

The activities of four key antioxidant enzymes, superoxide dismutase (SOD), catalase (CAT), ascorbate peroxidase (APX), and guaiacol peroxidase (GPX), were determined in ten-day-old *Chenopodium quinoa* seedlings to evaluate their response to oxidative stress induced by salinity. For enzyme extraction, 0.1 g of fresh plant tissue was homogenized in 1.5 mL of ice-cold extraction buffer (50 mM potassium phosphate, pH 7.0; 1 mM EDTA; and 1% polyvinylpyrrolidone [PVP]). The homogenate was centrifuged at 12,000 rpm for 15 min at 4 °C, and the supernatant was collected for subsequent enzymatic assays. The protein concentration was determined via the Bradford method [56]. All enzymatic analyses were performed at 25 °C, following adaptations of the protocols described by Sinha et al. [56].

SOD activity was evaluated via a method adapted from [57], which is based on the ability of the enzyme to inhibit the photochemical reduction of nitroblue tetrazolium (NBT). The reaction mixture (3.0 mL) included 50 mM potassium phosphate buffer (pH 7.8), 13 mM methionine, 75 µM NBT, 2 µM riboflavin, 100 µM EDTA, 50 mM sodium carbonate (pH 10.2), and 100 µL of enzymatic extract. The reaction was initiated by adding riboflavin and exposing the tubes to white fluorescent light (80–100 µmol photons/m^2^/s) for 15 min. The control group was kept in darkness. Formazan formation was evaluated by measuring the absorbance at 560 nm. SOD activity was expressed as the percentage inhibition of NBT reduction compared with that of the control without the enzymatic extract. One unit of enzyme activity was defined as the amount of enzyme required to inhibit NBT photoreduction by 50%.

CAT activity was determined spectrophotometrically by measuring the rate of H_2_O_2_ decomposition, according to the method described by Aebi [57]. The reaction system (2.0 mL) contained 50 mM potassium phosphate buffer (pH 7.0), 20 mM H_2_O_2_, and 100 µL of enzymatic extract. The reaction began upon the addition of the extract, and the decrease in absorbance at 240 nm was monitored every 30 s for 2 min via a UV-Vis spectrophotometer. Catalase activity, expressed as micromoles of H_2_O_2_ decomposed per minute per milligram of protein, was calculated from the linear decrease in absorbance via the molar extinction coefficient of H_2_O_2_ (ε = 36 mM^−1^ cm^−1^).

APX activity was measured via a modified version of the method described by Rekik et al. [58], which was optimized for *Chenopodium quinoa* seedlings. The 3.0 mL reaction mixture contained 50 mM potassium phosphate buffer (pH 7.0), 0.25 mM ascorbic acid, 5 mM H_2_O_2_, and 100 µL of enzymatic extract. The reaction was initiated by the addition of H_2_O_2_, and the decrease in absorbance was monitored at 290 nm for one minute via a UV-Vis spectrophotometer. The enzymatic activity was calculated via the molar extinction coefficient of ascorbate (ε = 2.8 mM^−1^ cm^−1^) and expressed as micromoles of ascorbate oxidized per minute per milligram of protein.

GPX activity was evaluated by monitoring the formation rate of tetraguaiacol, a product of the hydrogen peroxide-dependent oxidation of guaiacol. The total reaction volume (3.0 mL) consisted of 100 mM potassium phosphate buffer (pH 6.5), 15 mM guaiacol, 0.05% (*v*/*v*) H_2_O_2_, and 100 µL of enzymatic extract. The reaction was initiated by the addition of H_2_O_2_, and the production of the colored oxidized product was monitored by measuring the increase in absorbance at 470 nm for one minute with a UV-Vis spectrophotometer. GPX activity was calculated via the molar extinction coefficient of tetraguaiacol (ε = 26.6 mM^−1^ cm^−1^) and expressed as units per minute per milligram of protein (U min^−1^ mg^−1^ protein).

### 4.7. Statistical Analysis

Statistical analysis and procedures were performed via R 4.5.1 [59]. Linear mixed-effects models were fitted to evaluate the effects of salinity level, bacterial inoculation, and their interaction on germination, seedling development, and antioxidant enzyme activity. Model fitting was conducted via the *lme4 1.1-37* R package [60], employing restricted maximum likelihood (REML) to estimate variance components and accommodate both fixed and random effects in the model structure.

In the analysis framework, inoculation (I) and salinity level (S) were treated as fixed factors, whereas block (b) was included as a random effect to account for variation across repeated trials. The general form of the model is expressed as follows:$$Y_{ijk}=\mu +I_{i}+S_{j}+{\left(I\times S\right)}_{ij}+b_{k}+\varepsilon_{ijk}$$
where $Y_{ijk}$ denotes the observed value of the response variable corresponding to the i-th inoculation treatment, the j-th salinity level, and the k-th experimental block. Here, μ is the overall mean; $I_{i}$ is the fixed effect of the inoculation treatment; $S_{j}$ is the fixed effect of the salinity level; ${\left(I\times S\right)}_{ij}$ is the fixed effect of the interaction between inoculation and salinity; $b_{k}∼N\left(0,\sigma_{b}^{2}\right)$ represents the random effect of the block; and $\varepsilon_{ijk}∼N\left(0,\sigma^{2}\right)$ is the residual error term.

The model was implemented via the *lmer()* function from the *lme4 1.1-37* R package, with the following syntax formula:$$lmer(response\sim inoculation*salinity+\left(1|block\right),\;data=dataset)$$

When statistically significant main or interaction effects were detected, pairwise comparisons were performed via Tukey’s honest significant difference test, implemented through the *emmeans 1.11.2-8* R package [61]. The graphs were made with inti 0.6.8 R package [62].

The complete code and reproducible statistical analysis are available in Appendix A.

## 5. Conclusions

This study provides the first evaluation of *Azospirillum brasilense* as a plant growth-promoting bacterium (PGPB) in quinoa during germination under increasing salinity. BR-11001 and, particularly, BR-11002 retained high viability in NaCl-supplemented media, confirming their halotolerance, and their inoculation improved the germination percentage and cotyledon emergence across the 0–450 mM NaCl gradient, indicating enhanced physiological efficiency during early development. Inoculated seedlings, especially those inoculated with BR-11002, presented greater root and shoot growth as well as greater biomass under severe salinity stress, whereas antioxidant enzymes (SOD, CAT, APX, and GPX) were consistently stimulated, confirming a strengthened oxidative defense system. Taken together, these findings establish BR-11002 as a promising bioinoculant for enhancing the germination, seedling establishment, and stress resilience of *C. quinoa* in salt-affected soils. In addition to advancing the fundamental understanding of PGPB–quinoa interactions, this work provides a scientific basis for integrating halotolerant *A. brasilense* strains into sustainable agricultural systems to improve crop productivity and resilience in saline- and resource-limited environments.

## Figures and Tables

**Figure 1 plants-14-03829-f001:**
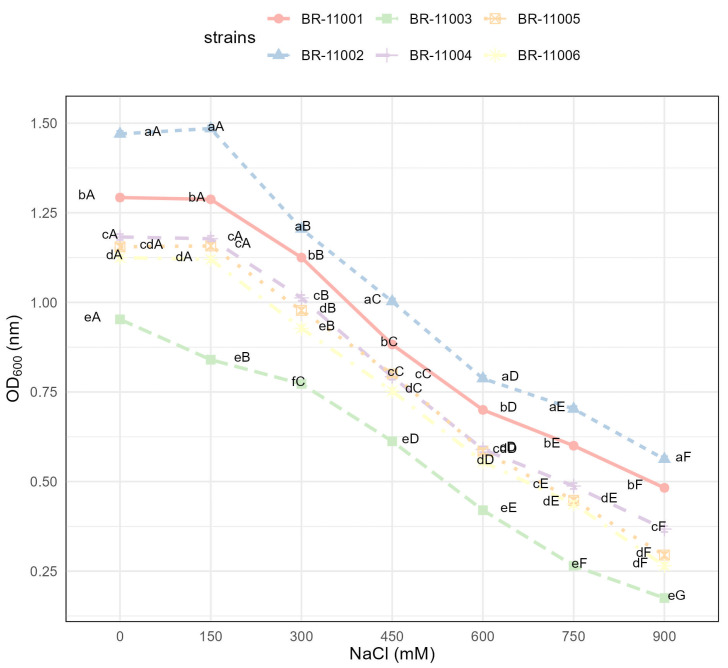
Optical density (OD_600_) of *Azospirillum brasilense* strains grown in Luria–Bertani media with NaCl concentrations ranging from 0 to 900 mM (*n* = 4). The error bars represent ± SEs. Lowercase letters indicate differences among strains within each salinity level; uppercase letters indicate differences among NaCl levels within each strain (Tukey’s HSD, *p* < 0.05).

**Figure 2 plants-14-03829-f002:**
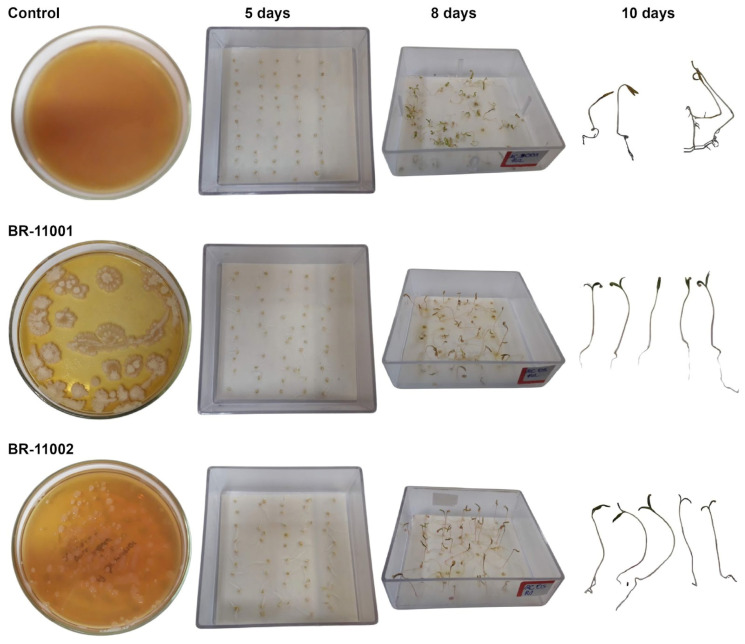
Visual comparison of *Chenopodium quinoa* germination and early seedling development under 300 mM NaCl at 5, 8, and 10 days after sowing in the uninoculated control, *Azospirillum brasilense* BR-11001, and BR-11002.

**Figure 3 plants-14-03829-f003:**
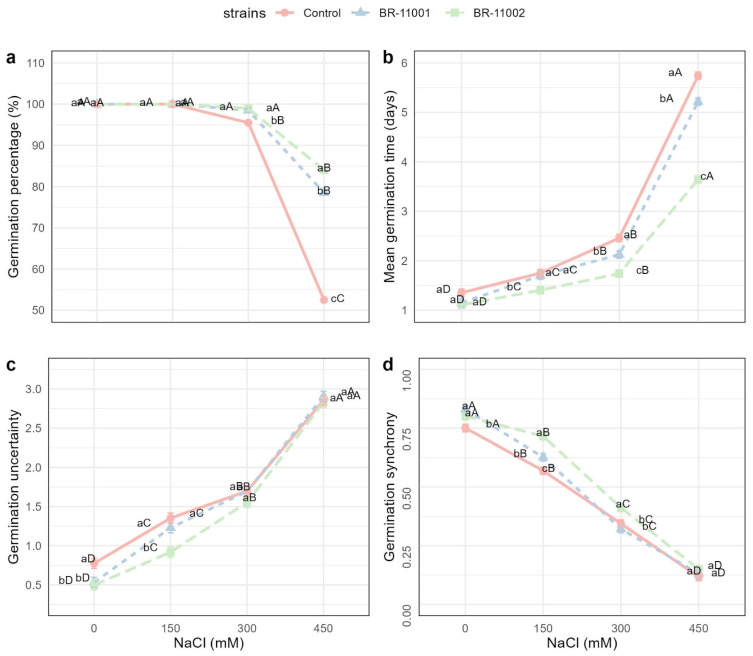
Germination responses of *C. quinoa* seeds inoculated with *A. brasilense* BR-11001 and BR-11002 under NaCl concentrations ranging from 0 to 450 mM (*n* = 4). (**a**) Germination percentage, (**b**) mean germination time, (**c**) germination uncertainty, and (**d**) germination synchrony. The error bars represent ± SEs. Lowercase letters indicate differences among strains within each salinity level; uppercase letters indicate differences among NaCl levels within each strain (Tukey’s HSD, *p* < 0.05).

**Figure 4 plants-14-03829-f004:**
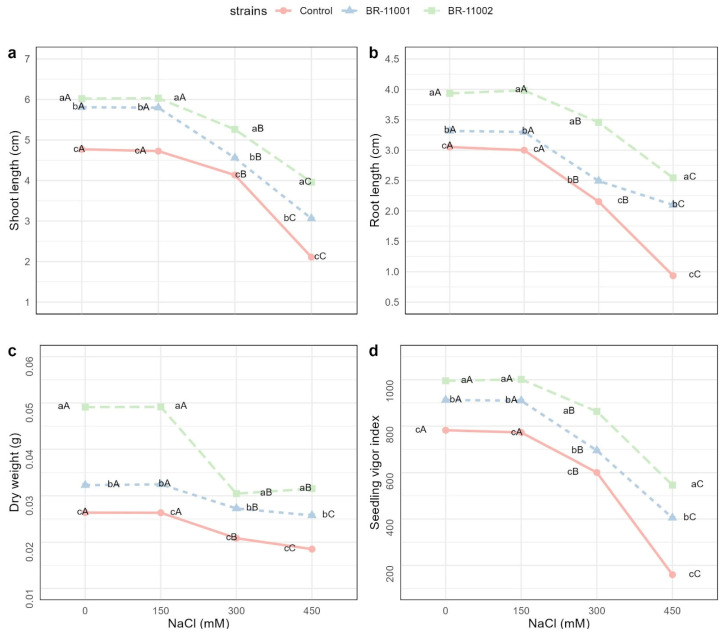
Traits of *Chenopodium quinoa* seedlings inoculated with *A. brasilense* BR-11001 and BR-11002 under NaCl stress. (**a**) Shoot length, (**b**) root length, (**c**) dry biomass, and (**d**) seedling vigor index. The data represent the means ± SEs (*n* = 4). Different lowercase letters denote significant differences among strains within each NaCl level; uppercase letters indicate differences among NaCl levels within each strain (Tukey’s HSD, *p* < 0.05).

**Figure 5 plants-14-03829-f005:**
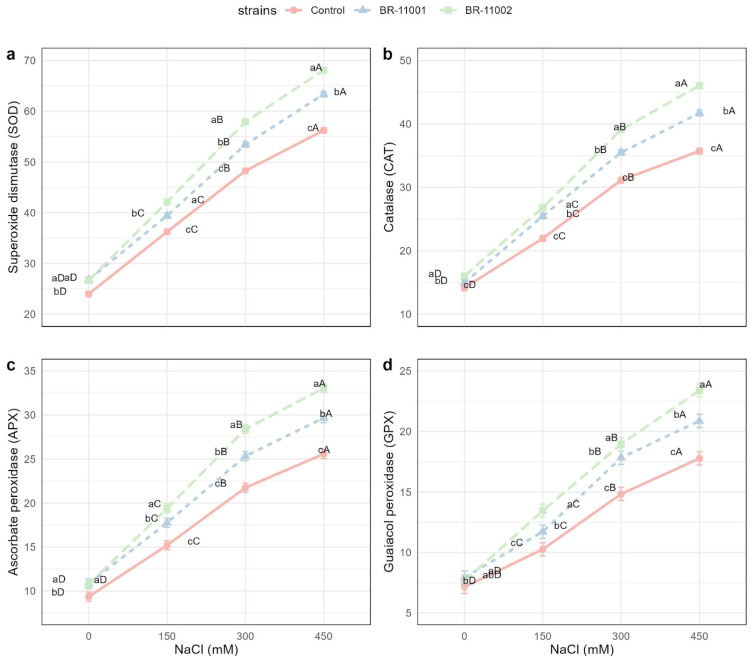
Antioxidant enzyme activity of *Chenopodium quinoa* seedlings subjected to salinity stress and inoculated with the *A. brasilense* strains BR-11001 and BR-11002. (**a**) Superoxide dismutase (SOD), (**b**) catalase (CAT), (**c**) ascorbate peroxidase (APX), and (**d**) guaiacol peroxidase (GPX). Enzyme activities are expressed as U min^−1^ mg^−1^ protein. The values represent the means ± SEs (*n* = 4). Different lowercase letters indicate significant differences among strains within each salinity level, and uppercase letters denote differences among salinity levels within each strain (Tukey’s HSD, *p* < 0.05).

**Figure 6 plants-14-03829-f006:**
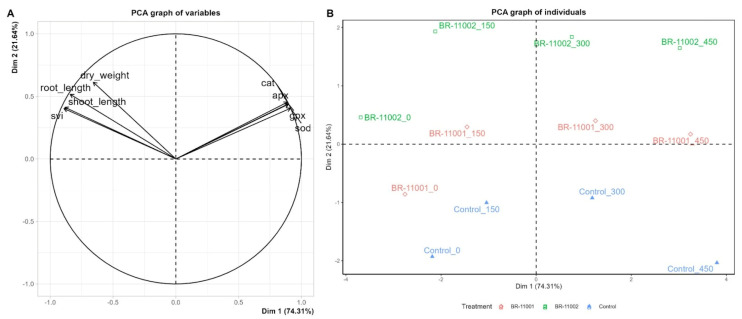
Principal component analysis (PCA) of the phenotypic and biochemical responses of quinoa (*Chenopodium quinoa*) seedlings inoculated with *Azospirillum brasilense* under salinity stress. (**A**) PCA of phenotypic variables: shoot length (SL, cm), root length (RL, cm), dry weight (DW, mg), and seedling vigor index (SVI, dimensionless); and biochemical variables: superoxide dismutase (SOD, U mg^−1^ protein), catalase (CAT, µmol H_2_O_2_ min^−1^ mg^−1^ protein), ascorbate peroxidase (APX, µmol ascorbate min^−1^ mg^−1^ protein), and guaiacol peroxidase (GPX, µmol tetraguaiacol min^−1^ mg^−1^ protein). (**B**) PCA of individuals represented by treatment combinations of bacterial strains (*Azospirillum brasilense* BR-11001, BR-11002 and the control) with different NaCl concentrations (0, 150, 300, and 450 mM).

## Data Availability

The data presented in this study are available in the GitHub repository: https://github.com/mila-ms/azospirillum_brasilense (accessed on 8 September 2025).

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
