# Peer review of "Azospirillum brasilense* as a Bioinoculant to Alleviate the Effects of Salinity on Quinoa Seed Germination"

_plants, 2025, doi:10.3390/plants14243829_

Round 1

Reviewer 1 Report

Comments and Suggestions for Authors

Abstract

The scientific names of the species should be written in italics: Chenopodium quinoa, Azospirillum brasilense.

Introduction

The main mechanism by which Azospirillum improves plant growth is through the production of phytohormones, primarily indole-3-acetic acid (IAA). The main pathway for IAA production is through the amino acid tryptophan (TRP). Describe the pathways for IAA production.

Among plant growth promoting bacteria, the genus Azospirillum is the most studied. Please describe what other bacteria are considered plant growth promoting.

Results and Discussion

The results showed that inoculation of salinity-stressed quinoa seeds with the endophyte A. brasilense improved germination at a dose of 300 mM NaCl; please discuss whether co-inoculation with other plant growth-promoting bacteria will improve quinoa seed germination.

Figures

Figures: Figure 1, Figure 3A and 3B, Figure 4A and 4B, Figure 5A and 5B are missing the legend for the X axis.

Author Response

Dear Reviewer, we sincerely thank you for your valuable comments and constructive feedback, which have helped us improve the clarity and quality of our manuscript. Below, we provide a point-by-point response to each of your suggestions. All changes have been incorporated into the revised version of the manuscript, with modifications highlighted in red.

The scientific names of the species should be written in italics: Chenopodium quinoa, Azospirillum brasilense.

We have revised the manuscript, and all scientific names are now consistently formatted in italics throughout, as recommended.

Introduction

The main mechanism by which Azospirillum improves plant growth is through the production of phytohormones, primarily indole-3-acetic acid (IAA). The main pathway for IAA production is through the amino acid tryptophan (TRP). Describe the pathways for IAA production.

A new paragraph has been added to the Introduction (lines 84–99 of the revised manuscript), describing in detail the tryptophan-dependent biosynthetic pathways for IAA production in Azospirillum, including the indole-3-pyruvic acid (IPyA), indole-3-acetamide (IAM), tryptamine (TAM), and indole-3-acetonitrile (IAN) pathways.

Among plant growth promoting bacteria, the genus Azospirillum is the most studied. Please describe what other bacteria are considered plant growth promoting.

A new paragraph has been incorporated in the Introduction (lines 73–83 of the revised manuscript), describing other genera widely recognized as plant growth-promoting bacteria (PGPB), including Rhizobium, Bradyrhizobium, Pseudomonas, Bacillus, Enterobacter, Klebsiella, and Burkholderia.

Results and Discussion

The results showed that inoculation of salinity-stressed quinoa seeds with the endophyte A. brasilense improved germination at a dose of 300 mM NaCl; please discuss whether co-inoculation with other plant growth-promoting bacteria will improve quinoa seed germination.

We have expanded the Discussion section to include a new paragraph (lines 309-325 in the revised manuscript) discussing the potential of co-inoculation strategies. Based on recent evidence (Yang et al., 2016; Yang et al., 2020), we highlight that co-inoculation of  Azospirillum with other halotolerant PGPB such as Bacillus or Burkholderia can enhance germination, seedling vigor, and stress resilience in quinoa under salinity stress.

Figures

Figures: Figure 1, Figure 3A and 3B, Figure 4A and 4B, Figure 5A and 5B are missing the legend for the X axis.

We appreciate this observation. To improve clarity and avoid redundancy, we have included the X-axis label only in the bottom panel of each figure group, as all panels share the same X-axis. We have now clarified this in the figure legends to ensure consistency and readability.

Reviewer 2 Report

Comments and Suggestions for Authors
  • Incorrect section order per journal guidelines: The "MATERIALS AND METHODS" section must be positioned as the ​second main section​ of the manuscript (current arrangement is non-compliant).

  • Inadequate articulation of research novelty: While the methodology states: "A total of six Azospirillum brasilense strains (BR-11001 to BR-11006) were provided by EMBRAPA, with documented salt tolerance and plant growth-promoting properties...", the research objective is superficially framed as "(i) to evaluate the growth potential of six bacterial strains under saline and non-saline media to assess viability and salt tolerance". ​This fails to explicitly highlight the study’s core innovative contribution.

  • Suspected numbering error: The subsection header "2.1.1. Bacterial growth under saline and non-saline conditions2.2." contains ​stray characters "2.2." likely resulting from formatting errors.

  • Required enhancement of statistical analysis: Figures 1, 3, 4, and 5 require ​supplemental ANOVA with significance designations​ for varying salt stress gradients (recommended: letter notation for significant differences).

  • Quantification recommendation for microbial growth promotion: Introduce ​correlation analysis between antioxidant enzyme activities and quinoa growth parameters​ (e.g., regression curves for SOD activity vs. plant height) to ​quantitatively evaluate microbial growth-promoting efficacy.

Author Response

Dear Reviewer, we sincerely thank you for your comments and feedback. Below, we provide a detailed response to your observation. All changes in the manuscript have been highlighted in red.

​Incorrect section order per journal guidelines: The "MATERIALS AND METHODS" section must be positioned as the ​second main section​ of the manuscript (current arrangement is non-compliant).

We appreciate this observation. However, according to the Plants journal’s official "Instructions for Authors”, the required structure for research manuscripts is: Abstract, Keywords, Introduction, Results, Discussion, Materials and Methods, and Conclusions (optional). Therefore, the current arrangement of our manuscript follows the journal’s prescribed format.

Reference: MDPI Plants - Instructions for Authors (Research Manuscript Sections) https://www.mdpi.com/journal/plants/instructions

​Inadequate articulation of research novelty: While the methodology states: "A total of six Azospirillum brasilense strains (BR-11001 to BR-11006) were provided by EMBRAPA, with documented salt tolerance and plant growth-promoting properties...", the research objective is superficially framed as "(i) to evaluate the growth potential of six bacterial strains under saline and non-saline media to assess viability and salt tolerance". ​This fails to explicitly highlight the study’s core innovative contribution.

We appreciate the feedback. The manuscript was restructured to clearly emphasize the study’s core contribution. In lines 73–120, we now specify that the role of Azospirillum brasilense as a plant growth-promoting bacterium (PGPB) in quinoa was evaluated, and to our knowledge, no prior studies have addressed its halotolerance-promoting effects during germination. Furthermore, the novelty was highlighted in the conclusion section (lines 545–546), underscoring that this is the first report on the use of A. brasilense to promote halotolerance in quinoa seedlings under salinity stress.

​Suspected numbering error: The subsection header "2.1.1. Bacterial growth under saline and non-saline conditions2.2." contains ​stray characters "2.2." likely resulting from formatting errors.

The subsection header has been corrected in the revised manuscript.

​Required enhancement of statistical analysis: Figures 1, 3, 4, and 5 require ​supplemental ANOVA with significance designations​ for varying salt stress gradients (recommended: letter notation for significant differences).

A factorial ANOVA was performed to evaluate the interaction between strains and salinity concentrations. The detailed outputs of these analyses are now provided in Supplementary Material 1 (indicated in lines 540–541 of the revised manuscript). In addition, the statistical analysis was improved and the figure legends for Figures 1, 3, 4, and 5 have been updated with the following description: “Different lowercase letters indicate significant differences among strains within each salinity level, and uppercase letters denote differences among salinity levels within each strain (Tukey’s HSD, p < 0.05).”

Quantification recommendation for microbial growth promotion: Introduce ​correlation analysis between antioxidant enzyme activities and quinoa growth parameters​ (e.g., regression curves for SOD activity vs. plant height) to ​quantitatively evaluate microbial growth-promoting efficacy.

To improve the interpretation of the relationship between biochemical variables and growth parameters, we added a section in the Results using a multivariate PCA analysis (Lines 263-288). The discussion of these interactions was also included in the revised Discussion (Lines 344-350).

Reviewer 3 Report

Comments and Suggestions for Authors

This paper lacks novelty. The authors primarily measure the phenotypic and biochemical responses of quinoa under salt stress to investigate the potential role of Azospirillum brasilense strains in alleviating salt stress. However, the mechanisms discussed are well-established plant responses to salinity stress and do not present any new scientific insights. Although the results indicate that these bacterial strains may alleviate salt stress in quinoa by enhancing antioxidant capacity, promoting seed germination, and improving seedling growth, these findings merely confirm known mechanisms rather than offering new discoveries.

Line 43, “Seed science” is too broad.

Line 43, “lant growth-promoting bacteria (PGPB)” should be corrected to “plant growth-promoting bacteria (PGPB)”.

Line 66-70, add references

Line 94, conditions 2.2?

Line 127, Why was 300 mM NaCl selected?

Line 195-213, The abbreviation of enzymes (SOD, CAT, APX, GPX) is introduced multiple times in the paragraph, but it would be clearer to define these abbreviations only once at first use and then consistently use the shortened form afterward.

Line 224-225, The mechanisms mentioned (such as enhanced germination, seedling growth, 224 and antioxidant responses) are common responses of plants to salt stress, and have been validated in other plants in many studies. The study merely repeats the same experiments on a different plant, lacking innovation.

Line 281-286, The description of the Azospirillum brasilense strains is detailed, but it would be helpful to include more context on how these strains were previously characterized (e.g., through laboratory assays or field studies). This would add credibility and clarity for readers who are less familiar with these strains.

Line 281-286, It might be worth specifying the functional or biochemical traits of each strain (if available), particularly how the different strains (BR-11001, BR-11002, etc.) vary in terms of salt tolerance or other key factors. This would support the rationale behind choosing these particular strains for the study.

Line 286, The authors mention that these strains have been used to alleviate salt stress in other plants in previous studies. Are the authors simply applying these strains to a different plant species in this experiment? If so, does this imply that similar experiments could be done with any plant species? If the aim of this study is to demonstrate the broad applicability of these strains across different plants, this should be clearly stated. However, if the study simply repeats existing research, the authors need to more clearly explain its contribution or further elucidate the novel findings of the experiment.

Lines 256-268, Although the observed phenotypic changes and biochemical responses may suggest that similar regulatory mechanisms are activated in quinoa, this is limited by the lack of direct gene expression data. The current inference is primarily based on known plant responses to similar salt stress, rather than on quinoa’s specific gene regulatory mechanisms.

Author Response

This paper lacks novelty. The authors primarily measure the phenotypic and biochemical responses of quinoa under salt stress to investigate the potential role of Azospirillum brasilense strains in alleviating salt stress. However, the mechanisms discussed are well-established plant responses to salinity stress and do not present any new scientific insights. Although the results indicate that these bacterial strains may alleviate salt stress in quinoa by enhancing antioxidant capacity, promoting seed germination, and improving seedling growth, these findings merely confirm known mechanisms rather than offering new discoveries.

We thank the reviewer for this observation. While it is correct that the general mechanisms of plant responses to salinity are well established, the novelty of our study lies in the evaluation of Chenopodium quinoa during its early developmental stages inoculated with halotolerant strains of Azospirillum brasilense. To our knowledge, this is the first comparative analysis of multiple A. brasilense isolates on quinoa germination, seedling growth, and antioxidant enzyme activity across a salinity gradient (0–450 mM NaCl). These aspects have been clarified in the revised Introduction (Lines 73–120) and further emphasized in the Conclusion (Lines 545–559).

Line 43, “Seed science” is too broad.

The keyword “Seed science” has been removed from the revised manuscript.

Line 43, “lant growth-promoting bacteria (PGPB)” should be corrected to “plant growth-promoting bacteria (PGPB)”.

The term has been corrected to “plant growth-promoting bacteria (PGPB)” in the revised manuscript.

Line 66-70, add references

References were added to support the statements regarding the role of plant growth- promoting bacteria (PGPB) in enhancing plant resilience under stress conditions.

Line 94, conditions 2.2?

The text has been revised for clarity.

Line 127, Why was 300 mM NaCl selected?

A justification has been added to the Materials and Methods section. Preliminary tests indicated that concentrations above 300 mM NaCl (e.g., 450 mM) severely inhibited quinoa germination, resulting in very low germination percentages, seedling necrosis, and morphological deformations. In contrast, 300 mM NaCl consistently produced a high germination percentage while still generating marked differences in seedling size and vigor compared with the control. For this reason, 300 mM was selected as the reference high-salinity treatment, as it provided a stringent but biologically meaningful level to assess the capacity of Azospirillum brasilense strains to alleviate salt stress during early growth (Lines 155-161).

Line 195-213, The abbreviation of enzymes (SOD, CAT, APX, GPX) is introduced multiple times in the paragraph, but it would be clearer to define these abbreviations only once at first use and then consistently use the shortened form afterward.

In the revised version of the manuscript, the antioxidant enzymes are now introduced in full only once at their first mention, followed by the abbreviation in parentheses. Specifically, superoxide dismutase (SOD), catalase (CAT), ascorbate peroxidase (APX), and glutathione peroxidase (GPX) are each defined at first use, and subsequently, only the abbreviations are applied consistently (Lines 238-253).

Line 224-225, The mechanisms mentioned (such as enhanced germination, seedling growth, 224 and antioxidant responses) are common responses of plants to salt stress, and have been validated in other plants in many studies. The study merely repeats the same experiments on a different plant, lacking innovation.

We appreciate your comment. While it is true that enhanced germination, seedling growth, and antioxidant responses are common plant reactions to salt stress, the novelty of our study lies in the evaluation of Azospirillum brasilense as a plant growth-promoting bacterium (PGPB) in Chenopodium quinoa under salinity during germination. To our knowledge, no previous study has reported the halotolerance-promoting effect of A. brasilense in quinoa at this critical stage. Brito et al. (2018) showed beneficial effects of A. brasilense on quinoa germination, but only under non-saline conditions. Similarly, Aslam et al. (2020) documented effects of rhizobacteria on quinoa early growth, and Yang et al. (2016, 2020) described other halotolerant bacteria in quinoa (This information was included in the Introduction to address the research gap at Lines 108-114). Therefore, our work provides the first evidence of A. brasilense-mediated halotolerance during quinoa germination, as also emphasized in our conclusion (Lines 544–549).

References

Brito, T. S., et al. (2018). Journal of Experimental Agriculture International, 1–9. https://doi.org/10.9734/JEAI/2018/39729

Aslam, M. U., et al. (2020). Communications in Soil Science and Plant Analysis, 51(7), 853–868. https://doi.org/10.1080/00103624.2020.1744634

Yang, A., et al. (2020). Plants, 9(6), 672. https://doi.org/10.3390/plants9060672

Yang, A., et al. (2016). Functional Plant Biology, 43(7), 632–642. https://doi.org/10.1071/fp15265

Line 281-286, The description of the Azospirillum brasilense strains is detailed, but it would be helpful to include more context on how these strains were previously characterized (e.g., through laboratory assays or field studies). This would add credibility and clarity for readers who are less familiar with these strains.

We appreciate this constructive suggestion. To address it, we expanded the introduction by adding two paragraphs that provide greater context. The first paragraph (Lines 73–83) introduces plant growth-promoting bacteria (PGPB) and their general mechanisms of action. And, the second paragraph (Lines 84–99) specifically describes Azospirillum brasilense, including its mechanisms and reported applications in non-leguminous crops. These additions aim to give readers unfamiliar with these strains a clearer understanding of their characterization and relevance, thereby strengthening the rationale of our study.

Line 281-286, It might be worth specifying the functional or biochemical traits of each strain (if available), particularly how the different strains (BR-11001, BR-11002, etc.) vary in terms of salt tolerance or other key factors. This would support the rationale behind choosing these particular strains for the study.

We thank you for the suggestion. In the revised manuscript in Material and Methods in Lines 381 – 410, we expanded the description to include previous studies on Azospirillum brasilense strains BR-11001, BR-11002, and BR-11005.

Line 286, The authors mention that these strains have been used to alleviate salt stress in other plants in previous studies. Are the authors simply applying these strains to a different plant species in this experiment? If so, does this imply that similar experiments could be done with any plant species? If the aim of this study is to demonstrate the broad applicability of these strains across different plants, this should be clearly stated. However, if the study simply repeats existing research, the authors need to more clearly explain its contribution or further elucidate the novel findings of the experiment.

It is correct that Azospirillum brasilense strains have been previously evaluated in other non-leguminous species  (Casanovas et al., 2002; E. Meyer et al., 2024; Fernando Shintate Galindo et al., 2020). Their ability to promote plant growth is linked to well-documented genetic and metabolic traits, including phytohormone synthesis, biological nitrogen fixation, siderophore production, and antioxidant enzyme stimulation, all of which contribute to mitigating salinity stress (A. Timofeeva et al., 2023). However, to the best of our knowledge, this is the first study to evaluate these halotolerant strains in quinoa during the germination stage, which represents a particularly vulnerable developmental phase. Quinoa is a facultative halophyte with unique mechanisms of salt tolerance, and our findings demonstrate that inoculation with strain BR-11002 significantly improves germination performance, seedling establishment, and antioxidant defense under high salinity. Thus, rather than a simple repetition, our work provides new insights into the application of A. brasilense in a pseudocereal of high nutritional and agronomic importance.

This research not only confirms the broad potential of these strains across different plant groups but also establishes a novel contribution by demonstrating their effectiveness in enhancing early-stage quinoa performance under saline conditions.

The explanation above was included in the main text at the Introduction to improve the research gap at Lines 73 - 120. And also highlighted in the Conclusion at Lines 544 - 549.

Reference:

  1. Timofeeva, Maria R Galyamova, & S. Sedykh. (2023). Plant Growth-Promoting Soil Bacteria: Nitrogen Fixation, Phosphate Solubilization, Siderophore Production, and Other Biological Activities. Plants. https://doi.org/10.3390/plants12244074

Casanovas, E. M., Barassi, C. A., & Sueldo, R. J. (2002). Azospiriflum Inoculation Mitigates Water Stress Effects in Maize Seedlings. Cereal Research Communications, 30(3–4), 343–350. https://doi.org/10.1007/BF03543428

Jalal, A., Filho, M.C.M.T., da Silva, E.C., da Silva Oliveira, C.E., Freitas, L.A., do Nascimento, V. (2022). Plant Growth-Promoting Bacteria and Nitrogen Fixing Bacteria: Sustainability of Non-legume Crops. In: Maheshwari, D.K., Dobhal, R., Dheeman, S. (eds) Nitrogen Fixing Bacteria: Sustainable Growth of Non-legumes. Microorganisms for Sustainability, vol 36. Springer, Singapore. https://doi.org/10.1007/978-981-19-4906-7_11

  1. Meyer, S. Stoffel, Anna Flávia Neri De Almeida, Juliana Do Amaral Scarsanella, A. S. Vieira, Bárbara Santos Ventura, Andressa Danielli Canei, J. Bortolini, Sergio Miana De Faria, C. R. Soares, & P. E. Lovato. (2024). Rhizophagus intraradices and Azospirillum brasilense im-prove growth of herbaceous plants and soil biological activity in revegetation of a recov-ering coal-mining area. Brazilian Journal of Microbiology. https://doi.org/10.1007/s42770-024-01390-2

Aizheng Yang, Yang, A., Saqib Saleem Akhtar, Akhtar, S. S., Qiang Fu, Fu, Q., Muhammad Naveed, Naveed, M., Shahid Iqbal, Shahid Iqbal, Iqbal, S., Thomas Roitsch, Roitsch, T., Sven‐Erik Jacobsen, & Jacobsen, S.-E. (2020). Burkholderia Phytofirmans PsJN Stimulate Growth and Yield of Quinoa under Salinity Stress. Plants, 9(6), 672. https://doi.org/10.3390/plants9060672

Fernando Shintate Galindo, Galindo, F. S., Paulo Pagliari, Pagliari, P. H., Salatiér Buzetti, Buzetti, S., Willian Lima Rodrigues, Rodrigues, W. L., José Mateus Kondo Santini, Santini, J. M. K., Eduardo Henrique Marcandalli Boleta, Boleta, E. H. M., Poliana Aparecida Leonel Rosa, Rosa, P. A. L., Thiago Assis Rodrigues Nogueira, Nogueira, T. A. R., Édson Lazarini, La-zarini, E., Marcelo Carvalho Minhoto Teixeira Filho, & Filho, M. C. M. T. (2020). Can sili-con applied to correct soil acidity in combination with Azospirillum brasilense inoculation improves nitrogen use efficiency in maize. PLOS ONE, 15(4). https://doi.org/10.1371/journal.pone.0230954

Lines 256-268, Although the observed phenotypic changes and biochemical responses may suggest that similar regulatory mechanisms are activated in quinoa, this is limited by the lack of direct gene expression data. The current inference is primarily based on known plant responses to similar salt stress, rather than on quinoa’s specific gene regulatory mechanisms.

We are thankful for this observation. Indeed, our study was primarily conducted at the phenotypic and biochemical levels, and we did not assess gene expression or molecular regulatory mechanisms. We fully agree that this limits the ability to directly link the observed responses to quinoa’s specific gene regulatory pathways. This point will be acknowledged in the revised manuscript, where we will include this limitation in the discussion section and highlight the need for future studies integrating molecular and transcriptomic approaches to provide a deeper mechanistic understanding of the halotolerance promoted by Azospirillum brasilense in quinoa. A section at Discussion was added entitle “Insights and limitations” at Lines 363-367 to include the research limitations and future steps.

Reviewer 4 Report

Comments and Suggestions for Authors

This study was conducted to explore the capacity of two halotolerant strains of Azospirillum brasilense (BR-11001 and BR-11002) to enhance salt stress tolerance in the quinoa cultivar 'BRS Piabiru'. The findings are meaningful. But major revision is necessary before going to possible publication in Plants. Below please find my detailed comments. I hope the authors can consider the following aspects when they revise the manuscript.

Abstract:

The first 1-2 sentences should address the current research gap.

P32: Pay attention to minor errors.

Introduction:

  1. The description of the study's innovation is insufficient and should be further improved.
  2. The rationale for investigating "the effects of A. brasilense strains on quinoa performance under saline stress" is unclear. Please revise and clarify.

Results:

  1. P94: "2.2"? Check for minor errors and review throughout the manuscript.
  2. P99, p< 0.xxx? Please add.
  3. Figure 1 should be more standardized—include the x-axis label, and avoid overlapping words.
  4. The descriptive results in Figure 2 should be moved to the Results section.
  5. P132-213: The Results section should focus on key findings. Besides, were there significant differences between the treatments? Please supplement this information.

Discussion:

  1. Avoid repetitive descriptions of results; instead, focus on explaining the mechanisms behind the findings.
  2. The Discussion section should be structured with subheadings for clarity.

Materials and Methods:

  1. P281-293: Some content would be more appropriate in the Introduction.

Conclusions:

  1. The Conclusions section is too brief. Please expand it by summarizing the key findings and highlighting the main contributions of this study.

The writing quality and language level of the paper need to be improved.

Author Response

This study was conducted to explore the capacity of two halotolerant strains of Azospirillum brasilense (BR-11001 and BR-11002) to enhance salt stress tolerance in the quinoa cultivar 'BRS Piabiru'. The findings are meaningful. But major revision is necessary before going to possible publication in Plants. Below please find my detailed comments. I hope the authors can consider the following aspects when they revise the manuscript.

Dear Reviewer, many thanks for your comments and feedback. We will provide a detailed response to each of your comments below. All changes in the manuscript were highlighted in red.

Abstract:

The first 1-2 sentences should address the current research gap.

We thank the Reviewer for this observation. In the revised abstract (Lines 27-31), the opening sentences were modified to emphasize the current knowledge gap. Specifically, we now highlight that although quinoa is recognized for its resilience to abiotic stresses, its early developmental stages - germination and seedling establishment - remain highly vulnerable to salinity, and that strategies to enhance tolerance during these stages are still limited. We also note that the role of halotolerant plant growth- promoting bacteria (PGPB) in quinoa has not been evaluated. This revision frames the research gap more clearly and provides the rationale for the present study.

P32: Pay attention to minor errors.

The manuscript was proofread, and all minor typographical, grammatical, and formatting errors were corrected throughout the text to improve clarity and readability.

Introduction:

  1. The description of the study's innovation is insufficient and should be further improved.

We thank the Reviewer for this observation. The Introduction section was revised to explicitly highlight the innovation of the present work (Lines 108–114). We now emphasize that this is the first systematic evaluation of halotolerant Azospirillum brasilense strains in quinoa, a crop of high nutritional and agronomic value, during germination and early seedling development under saline stress. This addition strengthens the novelty of the study and clarifies its unique contribution to the field. This information was also included in the Conclusion (544-558).

  1. The rationale for investigating "the effects of A. brasilense strains on quinoa performance under saline stress" is unclear. Please revise and clarify.

We thank the Reviewer for this valuable comment. The rationale was expanded in the Introduction (Lines 84–99) to explain that, although A. brasilense has been extensively studied in cereals and other non-leguminous crops, its effects on quinoa remain unexplored. Given that quinoa is a facultative halophyte with distinct salt tolerance mechanisms, testing A. brasilense provides an opportunity to identify bioinoculant strategies that can enhance germination, seedling establishment, and stress resilience under saline conditions. This revision clearly justifies the focus of the study.

Results:

  1. P94: "2.2"? Check for minor errors and review throughout the manuscript.

We corrected the reference to “2.2” and reviewed the manuscript to address minor errors.

  1. P99, p< 0.xxx? Please add.

We have added the corresponding p-values in the revised manuscript (Line 127 ).

  1. Figure 1 should be more standardized - include the x-axis label, and avoid overlapping words.

In the revised manuscript, Figure 1 was standardized by adding the x-axis label and adjusting the formatting to avoid overlapping words.

  1. The descriptive results in Figure 2 should be moved to the Results section.

The legend of Figure 2 was revised to include only essential information about the figure, while the descriptive results were moved to the Results section for clarity and consistency.

  1. P132-213: The Results section should focus on key findings. Besides, were there significant differences between the treatments? Please supplement this information.

The Results section has been revised to emphasize the key findings more clearly. In addition, we have supplemented the text with the outcomes of the statistical analysis, highlighting significant differences between treatments. Specifically, ANOVA followed by Tukey’s post-hoc test revealed significant variation among inoculated and control groups across salinity levels, which has now been explicitly reported in the revised Results.

Discussion:

  1. Avoid repetitive descriptions of results; instead, focus on explaining the mechanisms behind the findings.

Thank you for the suggestion. The Discussion was revised to reduce repetition and to emphasize the physiological and biochemical mechanisms.

  1. The Discussion section should be structured with subheadings for clarity.

The Discussion section has been revised and reorganized with subheadings to improve clarity, readability, and logical flow of the main findings.

Materials and Methods:

  1. P281-293: Some content would be more appropriate in the Introduction.

Thank you for your recommendation; however, in the Material and Methods we expanded the description of the strains used, as suggested by other reviewers (Lines 381-403).

Conclusions:

  1. The Conclusions section is too brief. Please expand it by summarizing the key findings and highlighting the main contributions of this study.

We thank the Reviewer for this observation. The Conclusions section was revised and restructured to directly respond to the study objectives (Lines 554–558) and expanded to provide a clearer summary of the main findings and contributions of this work.

The writing quality and language level of the paper need to be improved.

Thanks for the suggestion. To improve the manuscript’s clarity and style, we first used the Curie AI platform from Springer Nature (https://beta.springernature.com/pre-submission/writing-quality) to obtain detailed recommendations on writing quality. Subsequently, the revised version was carefully reviewed by a native English speaker to ensure accuracy, fluency, and consistency throughout the text.

Round 2

Reviewer 2 Report

Comments and Suggestions for Authors

I am grateful to the authors for responding to all my suggestions and comments and making the necessary corrections to the text of the manuscript. I have no more comments.

Reviewer 3 Report

Comments and Suggestions for Authors

The author revised the paper based on the reviewers’ comments.

Reviewer 4 Report

Comments and Suggestions for Authors

The authors have made revisions to the article according to the advices.

It can be accepted in the current version.